# Factors Associated with Mortality in Ontario Standardbred Racing: 2003–2015

**DOI:** 10.3390/ani11041028

**Published:** 2021-04-05

**Authors:** Peter Physick-Sheard, Amanda Avison, William Sears

**Affiliations:** 1Department of Population Medicine, University of Guelph, Guelph, ON N1G 2W1, Canada; wsears@uoguelph.ca; 2DVM Program, Ontario Veterinary College, University of Guelph, Guelph, ON N1G 2W1, Canada; Amanda.Jowett@hotmail.com

**Keywords:** sustainability, training, racing industry, work intensity, equine welfare, social license, risk factors

## Abstract

**Simple Summary:**

Racing provides employment and career engagement, is passionately pursued, and helps sustain our close relationship with horses, but it can also be associated with injury and losses. Fatalities occur on and off racetracks, involving welfare concerns, economic impact, and damage to racing’s public profile and social license. Musculoskeletal injury, the most visible loss, represents only one source and remains poorly understood, while for other losses and off-track mortality little is known. In 2003, the Province of Ontario, Canada introduced a registry for racehorse mortalities, providing opportunities to better understand losses and contributing factors. Following an earlier publication describing losses across all breeds, this paper presents analysis of standardbred mortality and relationships with routine management and competition. Results reveal that aspects of industry structure may contribute to mortality, and that the impact might be anticipated by close monitoring of a horse’s profile and performance. The immediate circumstances precipitating any specific fatality should be seen as separate from this underlying environmental liability. This has implications for how future research might be conducted and findings interpreted. It is hoped the present study can be used to decrease mortality and cumulative injury so as to reduce losses and strengthen societal support for racing.

**Abstract:**

Factors associated with mortality in standardbred racehorses were assessed through a retrospective annualized cohort study of all-cause mortality from 2003–2015 (*n* = 978) (identified in the Ontario Racehorse Death Registry). Race and qualifying data for official work-events were also gathered (1,778,330 work-events, 125,200 horse years). Multivariable logistic regression analysis revealed sex, age, and indices of workload and intensity and their interactions to be strongly associated with mortality. Track class, race versus qualifying performance, and work-event outcome (finish position, scratched, or failed to finish) also influenced mortality odds, which increased as performance slowed. Intense competition at higher performance levels and qualifying races at lower levels carried particularly high odds. Though occurring frequently, musculoskeletal injury was less frequent than all other presenting problems combined. Industry structure contributes to mortality through interaction between horse characteristics and the competition environment. This substrate may be amenable to management to minimize liability, but incident-specific triggers may represent chance factors and be relatively difficult to identify or control. Differentiating between substrate and trigger when studying specific clinical problems may provide greater clarity and yield in identifying underlying causes. Mortality may reflect a continuum of circumstances, cumulative impacts of which might be identified before a fatal event occurs.

## 1. Introduction

Injury and fatality in racehorses are issues of ongoing concern [1,2,3,4,5,6]. Extensive research has been performed for the thoroughbred, with primary focus on musculoskeletal injuries (MSI), the greatest source of loss. Many other factors that place horses at risk have been identified [7,8,9]. As a result, there is improved understanding of problem-specific mechanisms [10,11,12,13,14]. Since, for MSI, mortality may represent an end result of cumulative work-associated damage [10,11,12,15,16,17,18], environmental norms reflecting industry structure are likely to influence both morbidity and MSI-associated mortality. MSI is also not the only problem associated with mortality, and whatever the specific issue may be, almost all problems involving a racehorse arise in the context of its engagement in race preparation and racing.

“Industry structure” references the breeding, training, and campaigning of racehorses, track procedures/practices, race structures, performance demands, and workloads, as well as the imperatives, real and imagined, encountered in racing. It covers frequency and intensity of work, as well as characteristics of the work environment, from track surface and length to geographic location, regional track distribution, and horse movement, track and training center facilities, and prevailing economic circumstances. Finally, it encompasses the physiologic and psychologic dimensions of training and race intensity exercise [2,5]. These combine to create the environment in which racehorses work. Potential impacts of industry structure on losses has received limited attention. Underlying structures may be seen as immutable features of racing [1,19], yet such influences may play both direct and indirect roles in morbidity and mortality independently of specific mechanisms. Our management of racehorses creates the circumstances in which losses occur and can be viewed as forming an industry substrate within which specific triggers or discrete circumstances precipitate specific clinical episodes. It is this substrate that is the primary focus of this study.

Limited information exists on industry structure and morbidity for the standardbred racehorse and even less on mortality. Age and sex influences on morbidity have been examined with variable study designs and populations and with sometimes conflicting results, [20,21,22,23]. Geldings were found to have a higher incidence of lameness than females, as did 3-year-olds than 4-year-olds in one study [20], while another found no age effect and a lower incidence for geldings than stallions [21]. standardbreds were presented with pelvic fractures at younger ages than thoroughbreds or sport horses in a third study [23], while in a study of mortality, rate was high for very young horses, fell by age to age 5 years, then rose to be highest in mature horses [23]. In that same study, mortality rate was highest among stallions.

The influence of training on specific injuries has received some attention [24,25,26], but the influence of trainer has received insufficient study to draw firm conclusions. A driver effect has been noted [21,25], while workload and racing intensity/speed have also been incriminated [21,23,25], as have intense and high speed training predisposing to injury. Most studies have drawn on select populations yet how factors such as sex and age influence wastage remains unclear. Management and competition strategies and the resultant stresses they are likely to impose on the horse will vary with age and sex and with possibly fluctuating athletic ability and health status throughout a horse’s career. These are probably determinants of injury and trainer/owner response to those injuries. Improved understanding of underlying relationships could prove effective in reducing losses and addressing welfare concerns, helping build social license for racing and enhancing sustainability [2,27].

The Province of Ontario, Canada maintains a Racehorse Death Registry addressing all fatalities in standardbred, thoroughbred, and quarterhorse racing in the Province, on or off the track. Descriptive analysis of these data for 2003–2015 revealed mortality patterns to vary according to breed-specific profiles of age, sex, stage of career, and workload, as well as to reflect management and structural norms for each racing sector [23]. While high thoroughbred exercise mortality involved MSI, dying suddenly, and accidents, for example, rates for these complaints were low for the standardbred, where mortality involved a broader range of causes, was more frequently not exercise-associated, and tended to reflect the more extensive nature of standardbred preparation, training, and racing [28,29]. These observations imply significant impacts of differences in management.

The objective in the present study was to explore possible associations between industry structure and standardbred mortality. Mortalities were treated generically without differentiating by presenting problem and results thus apply to general mortality and do not describe problem-specific associations. Analysis addressed individual work events (race, qualifier) as the unit of interest and also horse-year (a single horse competing for a calendar year). Results expand upon conclusions drawn in descriptive analysis and quantify the impacts associated with industry structure, providing specific targets toward which regulatory strategies designed to reduce mortality might be directed.

## 2. Materials and Methods

The Ontario Racehorse Death Registry operates under Provincial Rules of Racing, which mandate that owners and trainers provide written notification to the regulator (Alcohol and Gaming Commission of Ontario, AGCO) within 2 days if a horse dies within 60 days of taking part in a race or qualifier, or of being entered to race or qualify. Penalties apply for failure to comply. Postmortem examination is mandatory where death takes place within 14 days and is otherwise at the discretion of the regulator. Horses withdrawn from a race (scratched) are also captured in the registry. Registry data for 2003–2015 inclusive were made available for the study by AGCO under a confidentiality agreement guaranteeing client anonymity.

Registry data for the study period contained 1713 cases of mortality, of which 978 involved standardbred horses. Three standardbred cases were eliminated because their deaths could not be confirmed, and two further horses were eliminated because of missing data, leaving 973 available for analysis. Registry data identified the animal, its age, sex and tattoo, the location, time and circumstances of death, and cause or suspected cause. Presenting complaint was that recorded by the submitting agent (most often the trainer or trainer’s agent) on the registry case submission form [23], and in most cases represented the diagnosis made by an attending veterinarian. Presenting complaints were consolidated into nine groups (Table 1), as previously described [23]. This process was aided by review of postmortem reports for all cases submitted to postmortem (55.21%). Mortality data were supplemented by performance data provided by Standardbred Canada in support of the study and describing details of officially recorded work, both race work-events and non-race work-events (predominantly qualifying races), from 1 January 2003 to 31 December 2015, inclusive for all standardbred horses competing in the Province of Ontario. Races and non-race events (i.e., qualifying races) are collectively referenced below as work-events.

A database was built containing all work-events, totaling 1,778,330 records covering 125,200 horse-years. Work-event variables available for use as independent variables in multivariable modelling are presented in Table 2, which presents definition, range, and data type for all variables used in the study. Three sexes (female, stallion, and gelding) were identified and for each event sex was that recorded for that work-event. The trackside terms “filly”, meaning young female, “colt”, meaning young intact male, and “horse”, meaning mature stallion, were not used. Model outcome was membership in the registry (dependent variable, DR). Derived variables were track class (TC, a surrogate measure of caliber of competition) and days between work-event and death for registry cases (DBD, Table 2). Three indices of cumulative officially recorded work (CMCAR, CMD, CMYR) were derived from race records and are also defined in Table 2. CMCAR included work-events from before 2003 for those horses racing before 2003.

Track class (TC) was determined by industry track classification, which considers facilities, location, intensity, and caliber of racing, track size, and speed rating. “A” (or premier) then “B” (signature tracks) were identified first, with the remainder (grassroots and regional) designated as “C” tracks. The work-event identified as associated with a mortality was the last work-event in which the horse participated prior to death. Temporal characteristics of this association have been previously described [23]. For horses captured in the registry because of entry for a race that took place after their death, the last work-event before death was defined as the work-event of interest and the subsequent work-event was deleted. Data analysis in this study was by calendar year. With the exception of CMCAR, parameters used represent annual statistics. Reference to effects taking place within or over a calendar year is by use of the terms “racing season” or “season”. “YEAR” refers to calendar year.

Main work-event outcome possibilities were; successfully completed (finishing positions 1–17, COMPLETED), scratched (horse withdrawn before the race started, SCRATCH), and failed to finish (did not finish the race, DNF). PROC FREQ (SAS 9.4^a^) was used to determine frequency of mortality for finishing positions within outcome COMPLETED and for outcomes SCRATCH and DNF. The continuous variable FPOS was then converted to categorical by grouping finishing positions with equivalent mortality rates. A second, categorical outcome variable, OUTC, was also created to combine all work-event outcome possibilities. Completed race categories for FPOS and OUTC were 1–5 (finish positions 1–5), 6–7 (finish positions 6–7), 8 (finish position 8), and 9–17 (finish positions 9–17). OUTC contained the additional outcomes SCRATCH and DNF (Table 2). Cumulative total for finishing positions 11–17 was 0.56% of records, representing run-off work-events for stakes races. Both FPOS and OUTC were retained in the database, but only one was used in any one model.

### Statistical Analysis

Logistic regression analysis was performed using PROC GENMOD or PROC GLIMMIX (SAS 9.4^a^), depending on preferred output, with a binomial response variable and logit link. Modelling proceeded by considering all main effects, then backward elimination was applied (preserving hierarchy). Terms taken out early were sequentially reintroduced to see whether they may have become significant after removing competing variables. Two-way interaction terms and quadratic effects were then introduced and examined. Stepwise addition and subtraction of terms was subsequently followed with retention of those significant at *p* < 0.05. Eventually, all variables and all possible two-way interactions were considered, regardless of biological plausibility. Second order effects were examined for all statistically significant continuous variables. Testing of three-way and four-way interactions was employed where indicated by combinations of significant two-way interactions. During construction of each model, model strength was assessed by monitoring type III tests of fixed effects (F-test *p*-value) to confirm the significance of each term retained in the model. Contrast estimates were constructed using PROC GLIMMIX (SAS 9.4^a^). Response was the binary variable DR (Death Registry membership). Significance was set at *p* ≤ 0.05 for all models except when DNF was modelled as the outcome, when a level of *p* ≤ 0.01 was employed because of group sizes and potential for identification of associations with questionable biological importance. Because of the number of horses and records involved, TATTOO (unique horse identifier) was not entered as a random variable in any model as it would have consumed all available degrees of freedom [30].

Results are expressed as odds and as odds ratios (OR) where comparisons are made. Odds and odds ratios are stated with their 95% confidence intervals and the significance level for the model estimate from which the OR was derived. Response (model outcome) was mortality (binary, membership in the Death Registry). In describing models, intercepts, estimates, standard errors, approximate 95% confidence intervals (CI), and p-values are presented for significant variables and all involved main effects. Odds ratios are presented together with their confidence intervals, where those intervals would provide a meaningful representation of population variation. Variables CMD, CMYR, and CMCAR were divided by 10 and variable YD was divided by 30 in most models to retain precision in significant estimates and to put results on a relevant scale and estimates and odds were adjusted accordingly. No attempts were made to determine risk or risk ratios in this modelling study.

Work-event outcomes DNF and SCRATCH were identified as highly influential in initial modelling. Separate mortality models were thus built with work-events stratified by the three main outcomes. An additional model was built with DNF as the outcome to assist with interpretation of the DNF mortality model. In this model, all work-events were included with the exception of scratches.

A final model with registry membership as the outcome was built in which the unit of interest was horse by year (horse-year model). Any horse racing in a year during the study period became a unit for that year, regardless of the number of work-events in which it participated. A horse could appear in consecutive years. For this model, SEX and variables DAY, CMD, CMYR, and CMCAR were those for each horse’s last work-event for each season.

PROC FREQ (SAS 9.4^a^) was used to examine categorical variables in support of grouping those with no significant difference in mortality association (*p* > 0.05), to reduce the number of degrees of freedom and model complexity. Estimates used in constructing contrasts were obtained by holding continuous variables at their mean and categorical variables at their referent levels.

## 3. Results

Mortality rates per 1000 work-events for outcomes COMPLETED, SCRATCH, or DNF are presented in Figure 1, by AGE and presenting problem (Data presented in Appendix A). Outcome group rates were 0.3369, 2.9377, and 30.7761 mortalities/1000 work-events, respectively, or 0.5441 overall. Each outcome group was modelled separately. Modelling of the full population (all outcomes) can be found in Appendix A.

### 3.1. Associations with Mortality for COMPLETED Outcomes

Results of logistic regression modelling by work-event outcome are shown in Table 3. Presenting complaints for COMPLETED outcomes are shown by AGE in Figure 1 and Appendix A. Mortality rate decreased with AGE, suggesting a survivor effect for older horses and highest mortality odds for young horses. SEX was influential, with stallions having consistently high odds of mortality. 

Effect of an AGE×SEX interaction was greatest for the difference between young geldings (low liability) and young stallions (high liability), with direction of this difference reversing with increasing AGE (Figure 2A). Mortality rate was relatively high for work-events involving young stallions regardless of the presenting problem; it fell to a low by age 5, then increased again (Figure 2B, data presented in Appendix A). Females and geldings both experienced increasing mortality odds as they aged when compared with stallions (Figure 2A).

A three-way interaction, AGE×SEX×CMD (significant for the gelding, *p* = 0.004) revealed AGE to be associated with increasing mortality odds for all sexes when CMD (cumulative days raced) was high (Figure 3). For work-events involving geldings, odds of mortality were low at low CMD and increased progressively with AGE and time racing. For female and especially stallion work-events with low CMD, odds were initially high and fell with AGE, then rose again (Figure 3). The pattern for young stallions suggests a potential impact of behavior and experience. The proportion of total work-events involving females fell by age from 47.95% for 2-year-olds to 5.74% for 12-year-olds, while stallion work-events fell from 13.46% to 9.94%. Work-events involving geldings rose from 38.95% for 2-year-olds to 84.31% for 12-year-olds.

Mortality odds in a qualifier (START = N) were 1.396 (1.079–1.807, *p* = 0.01, Table 3) times greater than in a race work-event after controlling for all other effects. There were 76 mortalities temporally associated with COMPLETED qualifying races for a mortality rate of 0.398/1000. YEAR was significant, with decreasing odds through the study period. Mortality rate also fell as track class rose from “C” to “A” (Table 4), with horses qualifying at “C” tracks having higher odds of mortality. Horses finishing in the first 5 positions had the lowest odds of mortality, with odds rising progressively as horses finished further back (see Figure 4, which also presents DNF outcomes for comparison). Gait did not enter the model.

### 3.2. Associations with Mortality for DNF Outcomes

Mortality was associated with AGE, SEX, GAIT, START, and TC (track class) as main effects, with no significant interactions or second-order effects (Table 3). Chances of dying in association with DNF increased with AGE by approximately 20% each year (Figure 1 and Appendix A), whereas overall odds of DNF fell with AGE at *p* = 0.02 (Appendix A). Females and geldings were less likely than stallions to experience mortality in a DNF work-event, though there was no SEX effect on the odds of failure to finish. Odds of mortality in DNF pacing work-events were almost twice those in DNF trotting work-events, though GAIT did not influence odds of DNF. Mortality odds with DNF in races were almost four times those for non-race work-events, though DNF was significantly more likely to take place in non-race work-events (Appendix A). DNF was more likely to be associated with mortality on “A” than “C” tracks. When DNF mortality rate was stratified by START, race mortality was by far the highest at “A” tracks (71.759/1000 work-events), while non-race rate was highest at “C” tracks (Table 4).

Presenting complaints for mortality in work-events with DNF outcomes (Figure 1, Appendix A) were predominantly musculoskeletal, dying suddenly, and accidents (total 96.13%), with mortality occurring in only 3.08% (207/6726) of total DNF outcomes. Horses dying in association with DNF were DNF on that one occasion, whereas for DNF outcomes not associated with mortality, individual horses were DNF during their careers from 1–11 times. Of 973 standardbred mortalities, 21.27% (207) occurred in association with DNF, with 84.80% (173) of these being associated with live racing. Of these 173 mortalities, 142 or 82.08% occurred within 24 h of the work-event, and 121 or 69.94% occurred on the same day. Presenting complaint rates (deaths per 1000 work-events) also varied by TC for DNF mortality (Table 4). Rates were highest for musculoskeletal disease, with little difference between track classes. Rate for dying suddenly was highest on “A” and lowest on “C” tracks, with accidents following a similar pattern. DNF iatrogenic mortality (fatal outcomes of procedures and treatments applied by managers, such as injection reactions) was highest on “C” tracks.

### 3.3. Associations with Mortality for SCRATCH Outcomes

Only SEX, YEAR, and the interaction CMCAR×SEX were significantly associated with mortality in SCRATCH work-events (Table 3). A scratch was 3 times more likely to be associated with mortality for a stallion than for females or geldings. The interaction CMCAR×SEX was only significant for females versus stallions, but when contrasts between SEX and CMCAR subgroups were examined (Appendix A), mortality was significantly higher for stallions than for mares and geldings at lower levels of CMCAR with the difference falling as CMCAR increased. Mortality odds in association with a SCRATCH work-event increased for females and decreased for geldings and stallions with increasing CMCAR. By CMCAR = 140, female mortality odds exceeded stallion odds. Odds for geldings remained below those for stallions despite odds for both decreasing with increasing cumulative career work-events. The odds of mortality in association with SCRATCH decreased progressively throughout the study period (Table 3). Differences in scratch-associated mortality between track classes (Table 3), did not reach statistical significance at *p* = 0.05.

Distribution of presenting complaints for SCRATCH-associated mortalities was similar to that for COMPLETED work-events, though with fewer musculoskeletal complaints and more frequent medical complaints and colic (Figure 1, Appendix A). Overall mortality rate for SCRATCH work-events was 2.9377/1000 (191/65,105 work-events), which was higher than COMPLETED work-events. SCRATCH iatrogenic and dying suddenly mortality rates were highest at “A” tracks (Table 4). Mortality was exercise-associated in 41/191 deaths (21%). In all instances of this outcome mortality took place subsequent to the horse being scratched from a work-event. Death occurred on the same day for 30.25% of 183 horses for which information was available and within 24 h in 44.81%. Of the total SCRATCH outcomes, 99.71% did not involve mortality. Information is not retained by the industry on scratches for qualifying races and all data are thus for race entries.

### 3.4. Modelling of Mortality by Horse-Year

Mortality odds were significantly lower for females and geldings than stallions (Table 5), but increased with AGE for geldings, exceeding stallion odds by age 10, while odds fell with AGE for stallions (Appendix A). AGE × YEAR interaction revealed mortality odds to fall by YEAR and rise with AGE, with the impact of AGE diminishing rapidly over the study period (Figure 5A). This mirrored the same interaction identified with work-event as unit of interest (Figure 5B).

Career indices were highly significant. Increasing annual and career work-events were associated with increasing odds of mortality independently of AGE, indicating a contribution of cumulative workload and intensity. The interaction between AGE × CMYR (Figure 6A) revealed exponential relationships, with mortality odds increasing with increasing cumulative year work-events progressively more rapidly with increasing AGE. A significant second-order effect was also identified for cumulative year work-events. Eliminating DNF outcomes from horse-year analysis did not change these relationships. The interaction AGE×CMCAR (Figure 6B) also indicated rising odds with increasing cumulative career work-events, but the relationship with AGE was reversed compared with cumulative year work-events, so that the interaction was most marked for young and least apparent for older horses. Curves are theoretical and show trajectories revealed by modelling; young horses would not achieve the career totals indicated. Results suggest high career starts for young horses are associated with high mortality whereas this is not the effect for horses with more extended careers. The effect for older horses is likely to include a survivor effect.

Increasing CMD (cumulative days raced) was associated with decreasing mortality odds as a main effect at horse level, while the interaction TC×CMD revealed odds to fall more rapidly with increasing CMD at “B” than “C” tracks (Appendix A). Horse-year odds at “B” tracks exceeded that for all others for very low CMD, but thereafter were highest at “C” tracks, though with a similar trajectory. These differences could indicate a combination of a training and survivor effect, whereby less robust horses progressively leave the population, and that competitive pressures are initially high on “B” tracks.

## 4. Discussion

Limited data exists on standardbred mortality. Previous studies have addressed morbidity or longevity and career profiles [31,32,33,34,35], and factors predisposing to lameness [20,21] but not mortality. It has been suggested standardbreds have a low work-event rate of injury because they race less intensely than other breeds [19]. Present results are not inconsistent with this interpretation; a race start carries higher odds of mortality for a thoroughbred than a Standardbred [23]. Most current information is for the thoroughbred and focuses on MSI during flat racing [9,36,37], with few reports on more general mortality [38] or morbidity [39]. A very incomplete picture of losses is acquired if only MSI are considered [39].

Age and sex are highly influential in injury, survival, and career length for racehorses [9,18,20,23,31,32,34,37,40,41,42,43,44], and the effects are complexly intertwined with management and industry structure. Comparison with previous studies is difficult because of wide variation in study group, selection criteria, reference population, and study design, and because of a dearth of studies addressing the racing standardbred. Moreover, the age range over which populations have been studied is often limited, interactions have received insufficient attention, and group sizes have been small. In general, the literature indicates a gradual increase in injury as horses age, both in thoroughbreds [9] and standardbreds [23], with the effect moderated by AGE×SEX interactions, and higher susceptibility for early career animals, as observed here. It is reasonable to expect since MSI is the most common contributing cause to mortality [7,8,9], that there would be a concomitant MSI-related influence on mortality, though most studies do not address this directly. The relative importance of other causes of mortality, as demonstrated here, has not been previously investigated.

The relationship with sex seems to be particularly strongly influenced by the variations in study design noted above, particularly the age range of studied horses. For example, in the present study odds were particularly high for young intact males but low for geldings, which showed much higher mortality as they aged. Odds for females were intermediate. In contrast, in a recent meta-analysis of musculoskeletal injuries in thoroughbred horses, despite a high overall rate for intact males, results varied widely for the effect of sex [9]. The distribution of odds ratios observed here indicates that this AGE×SEX interaction may not have been noted with a smaller population of study subjects with a narrow range of ages. In studies identifying a sex effect on injury frequency for thoroughbreds and for standardbreds results thus tend to be conflicting [20,21,23,34,41,42]. An item of importance to consider for the present study is that this analysis was performed on a complete population and with minimal missing data, that is, no population selection was employed, and results are parameters and not estimates. The applicability of the results to other populations, however, is undetermined.

Associations with sex noted here include whether or not a stallion was castrated, with young intact horses carrying significantly higher mortality odds than geldings. These effects could reflect multiple factors, including biology, wear and tear, and pathophysiology, as well as genetic contributions [45]. Recent evidence suggests a sex-differentiated genetic predisposition to fracture that also relates to superior performance ability [46], implying selection for speed may simultaneously select for fracture predisposition, as may speed itself [15]. Imperatives that drive selection and work stresses reflect industry practices and expectations, and these same forces also influence how age and sex are associated with mortality.

Findings suggest a major behavioral component to mortality and by inference to morbidity, and that youth, inexperience, and associated behaviors might be considered as possible primary mortality contributors. There are challenges involved in working with young racehorses [47,48], which face many sources of stress in the racing environment [49,50,51,52]. Aggressiveness and vitality in young, intact horses may be seen to confer competitive advantage, while anticipated loss of these attributes may be seen as one reason to delay castration of stallions. Without anticipatory behavior modification; however, this perspective may be associated with greater cost than benefit.

Higher mortality among young stallions, and to a lesser extent, females, raises the possibility otherwise promising horses may be lost early in training. Significantly lower mortality in 2-year-old geldings suggests strategies designed to blunt behavioral responses to introduction to early training, the track, and race intensity exercise by application of learning theory could yield benefits [48,53,54], and might not need to involve castration. Benefits could include reduced injuries and mortality, increased ease of training and general management, and opportunity for horses to express their full genetic potential [50,51,55,56]. The approach would also promote a reduction in human injuries [57,58]. Targets would be to diminish aggression and response to conspecifics, reduce sexual behaviors, familiarize horses with environmental change, increase routine contact with other horses, reduce isolated stall time, and simultaneously improve overall welfare. Such strategies could reduce stress levels for racehorses throughout their time in training, and by doing so, potentially enable them to better handle the inevitable acute physiologic and psychologic stress that is likely to result from intense exercise, whether training or race. Strategies would need to be applied starting at the breeding farm and could take several seasons to fully implement. This is an animal welfare issue as well as being of practical and economic significance [59].

Our performance indices emphasized cumulative work and examined their relationship to general mortality odds. Effects depended on unit of interest, while the relationship between work, age and career stage is complex in the horse [18,60,61]. At the work-event level, increasing annual work duration and frequency were associated with decreasing mortality odds for young horses, indicating horses were at greatest risk when first entering training [62], and possibly response to training for successful horses. For older horses, the relationship reversed, with increase in annual work increasing work-event mortality odds. In contrast, at horse-year level young horses with a high number of season and career starts had greatly increased mortality odds and decreasing tolerance of work intensity with age, both consistent with cumulative wear and tear. This may equate to superior ability to tolerate work in some older, proven horses, but may also indicate older horses benefitted from a less intense career. While these relationships emphasize the importance of age, it is difficult to separate them from a survivor effect, whereby withdrawal of less robust horses leaves a progressively more work-tolerant population. standardbred horses racing today have a shorter and more intense career than was the case in the 1970s, (unpublished observations) [32], suggesting we may be moving in the wrong direction. Findings have implications for both animal welfare and resource utilization.

In the Province of Ontario, 16 standardbred tracks were active during the study period, classified here as “A”, “B”, and “C”, with significant differences in mortality rate and associations. Mortality on “A” tracks in DNF outcomes with catastrophic breakdowns and higher rates of sudden death and accidents during races in older horses suggest speed and intensity of competition as significant factors in otherwise well-prepared, experienced horses. Young horse mortality at “B” tracks suggests pressure to perform and pursuit of targets and economic return on what may represent a proving ground. High non-race mortality rate at “C” tracks suggests attempts to re-qualify horses with deteriorating performance, plus possibly higher training mortality. These observations and associations with iatrogenic mortality raise questions of quality of care and career planning and provide focus for preventive strategies such as enhanced monitoring and pre-race examination. They also concern structural elements of the industry for which there may be better alternatives that place horses’ well-being on an equal footing with issues of economic return and survival. It would be informative to determine what proportion of such horses previously competed at higher levels. For a horse to move down in competitive class and continue competing until it is no longer able to do so would raise serious welfare concerns.

Factors influencing race outcome appear closely related to those influencing odds of mortality, regardless of presenting problem. A similar finding was identified in New Zealand thoroughbreds in relation to odds of MSI subsequent to absences from training [63]. A fatality may represent the endpoint of a continuum of liabilities that reflect how we train, maintain and campaign racehorses, plus basic horse characteristics, rather than an event solely attributable to a discrete cause or trigger. Present analysis suggests the substrate represented by the competition environment and fundamental characteristics of the horse should be seen as being of primary importance, with the circumstances triggering a clinical episode being secondary. Horses are the industry’s primary resource and are costly to prepare and maintain, while ideally, and recognizing obvious issues of welfare and industry social license, compromised horses would be withdrawn rather than experience fatal injury/breakdown in competition. It may be most conducive to industry success to cautiously optimize the number of quality earning opportunities and distribute costs over a long career. This requires planning, consistency, moderation, and an emphasis on longevity, informed management and continuous monitoring as basic operating strategies. Horses at risk of mortality could perhaps be identified by tools such as performance profiling [64], and mortality thus prevented. The same approach could be applied to identify horses most in need of withdrawal from competition.

### Limitations

Some mortality may have taken place without close work association or the association may have been coincidental. Future studies could stratify data on the basis of exercise association and use random interviews to explore the role of non-work factors. CMD represents the interval between first track appearance in a season and day of the current work-event. Most standardbred horses race continuously once started, but some may have had within-season absences. During the study period the industry was under intense pressure due to changes in government programs, and experienced significant contraction. This may have influenced decision-making concerning treatment versus euthanasia, which was the dominant immediate cause of death in this dataset. Such decisions may be influenced by humane concerns, economics, feasibility of other career options, and prognosis for future performance. Uncertainty is introduced into the data since the basis of these decisions is unknown. No information is gathered by the industry on scratch outcomes for qualifying races, and no comparison could be made between races and qualifying work-events in the SCRATCH model.

The definition of mortality used here was constrained by terms of the Death Registry. Losses occurring outside the 60-day window and among horses in early training, not racing, or used for breeding were not captured. Registry data do not address morbidity and findings do not present a comprehensive assessment of wastage. Annualized mortality rates based on work-events are susceptible to population dynamics and are influenced by changes in number of horses dying (numerator), and reference population size (denominator). For young horses, rates can be biased downward by new horses entering the population moderated by relatively low number of work-events by season. For older horses, rates can be biased upward because the population is shrinking, moderated by a relatively high number of work-events per horse. This study modelled probabilities, odds, and odds ratios—not risk ratios. Risk can be cautiously inferred from the results obtained, however, since the incidence of mortality was low in most instances [65]. Information on other factors that could have contributed to mortality, such as intercurrent disease, local weather, race strategy, details of training regimens, and clinical histories, were not available. Such information would allow more granular analysis by which the role of substrate and triggers might be more thoroughly examined.

## 5. Conclusions

Mortality in the Ontario standardbred racehorse has a broad association with frequency, intensity, and quality of work, as well as performance history, age, and sex. A relationship with structural elements of the industry such as track class and the prosecution of racing provide additional parameters by which horses at risk might be identified. Mortality is not an inevitable outcome of racing and may represent the endpoint of a continuum of influences whose effects might be anticipated. Circumstances influencing mortality may reflect interaction between a substrate consisting of intrinsic horse characteristics and the competitive environment, and problem-specific triggers by which the combined effect of stressors and chance events precipitate a specific clinical episode. Triggers may be difficult to identify, enumerate, control or foresee, and may masquerade as seemingly benign circumstances. Substrate factors, once recognized, might be manipulated, managed or pre-empted to minimize liability to adverse outcomes when a trigger is encountered. Circumstances that appear to carry particularly high odds, such as intense competition and frequent need to requalify, should receive particularly close attention. Analysis suggests that at these levels, while recognizing we have much to learn concerning specific triggers, we may already have significant components of the information required to have an impact on mortality and possibly morbidity as well.

## Figures and Tables

**Figure 1 animals-11-01028-f001:**
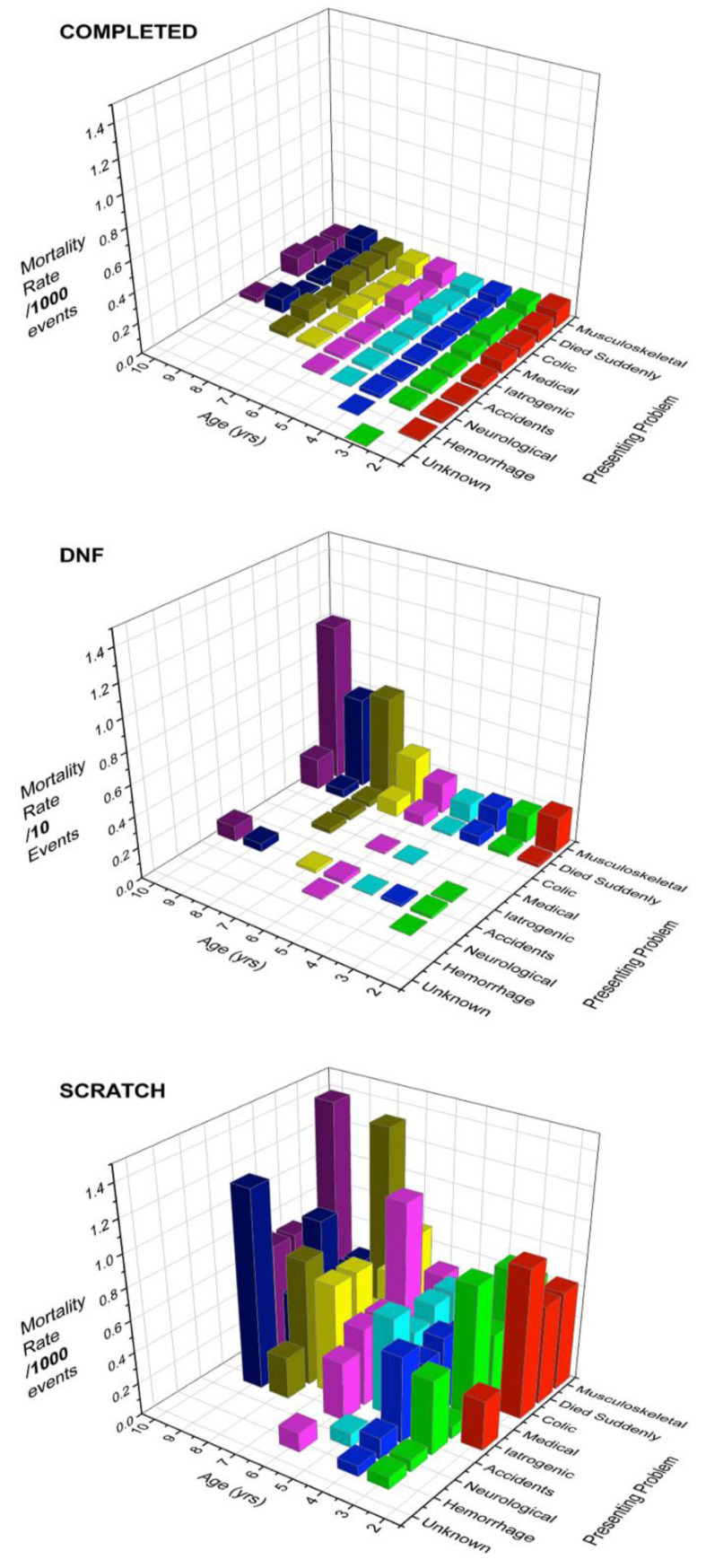
Mortality rates/1000 work-events (Note: /10 work-events for outcome DNF) for standardbred horses racing in the Province of Ontario from 2003–2015. Data are drawn from the Ontario Death Registry, unit of interest work-event. (*n* = COMPLETED 1,706,499, SCRATCH 65,105, DNF 6726). Rates are stratified by race outcome, AGE, and presenting problem, showing raw data rates without any adjustment for the influence of other factors. Results for ages above 10 years are not presented because group sizes were small and rates erratic. The problem category “unknown” reflects a single horse with no available data. Rates are low for COMPLETED outcomes but rise for SCRATCH outcomes and are highest for DNF outcomes. Note also that distribution of presenting problems is broad for COMPLETED and SCRATCH outcomes, but for DNF outcomes, musculoskeletal injury is by far the most frequent problem regardless of age, with dying suddenly and accidents being the next most frequent problems. Data for this Figure are presented in Appendix A.

**Figure 2 animals-11-01028-f002:**
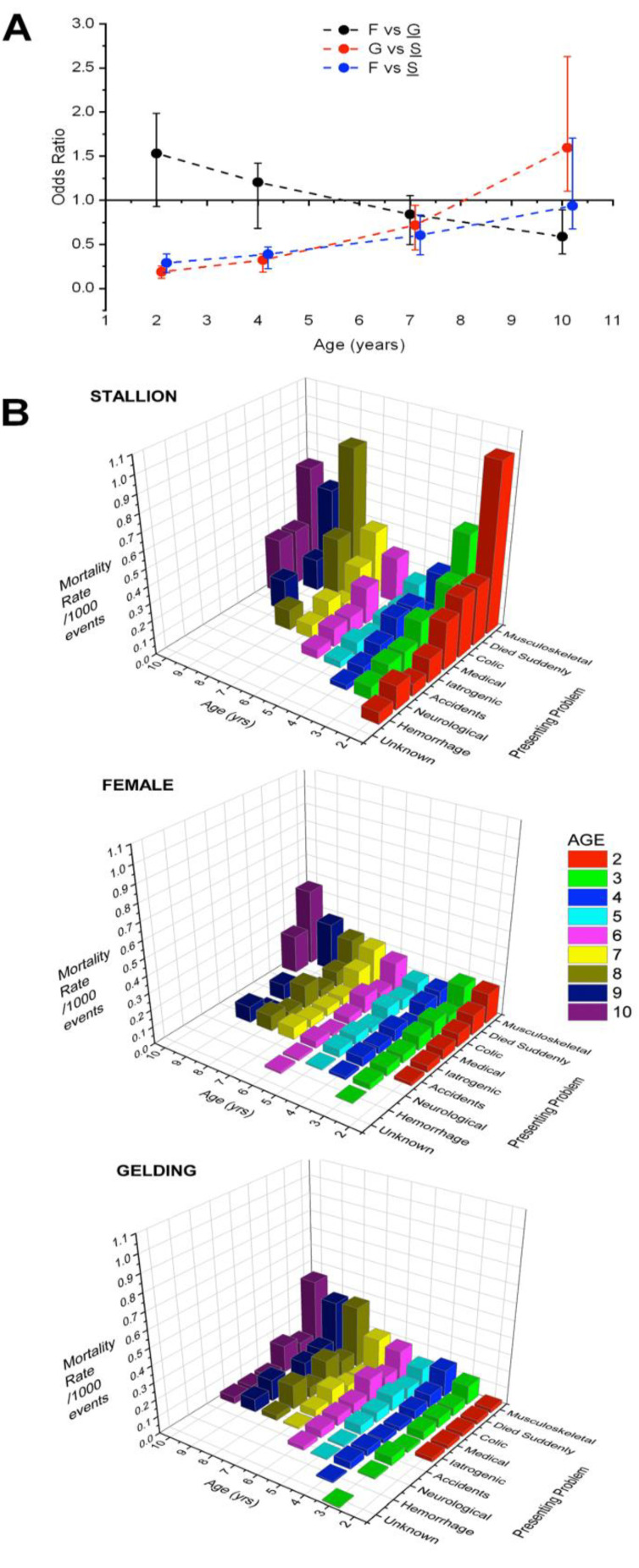
Mortality data from the Ontario Death Registry for work-events involving standardbred horses racing in the Province of Ontario in the period 2003–2015. Mortality by AGE (years) and SEX and describing a significant interaction identified through logistic regression analysis (outcome—binary response Registry membership, unit of interest—work-event). (**A**) Odds ratios for mortality and their 95% confidence intervals for the mean, comparing SEX and AGE group pairs. Referent SEX for each comparison is underlined in the figure legend. Thus, geldings have lower mortality rate than stallions at all ages except 10 years. F: female; G: gelding; S: stallion. Population-all work-events (*n* = 1,778,330). Data points are offset horizontally for clarity. (**B**) Mortality rates/1000 work-events stratified by SEX, AGE, and presenting problem. Data show raw rates without adjustment for the influence of other factors. Results for AGE > 10 years are not presented because group sizes were small and rates erratic. The problem “unknown” reflects a single horse for which no data were available. Appendix A shows data for this graph.

**Figure 3 animals-11-01028-f003:**
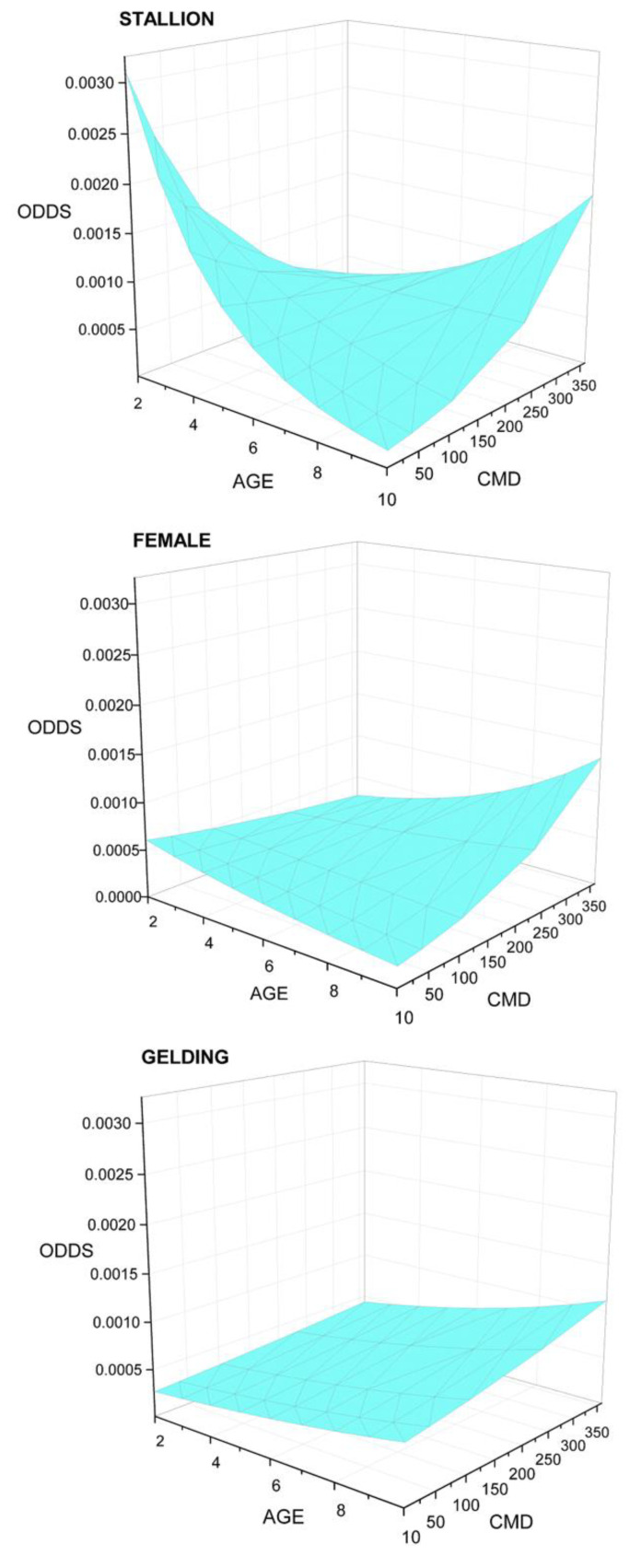
Odds of mortality by AGE, cumulative days raced in the season (CMD), and SEX for standardbred horses racing in the Province of Ontario from 2003–2015 and describing an AGE×CMD×SEX interaction identified in logistic regression analysis of mortality data from the Ontario Death Registry. Unit of interest for this analysis is work-event, population is work-events that finished normally (COMPLETED, *n* = 1,706,499). Three-dimensional response surfaces describe the interaction between AGE, CMD, and mortality odds for each sex group. Mortality patterns differ by SEX when all other factors are held constant. Patterns for stallions and females are similar but differ in degree, while pattern for the gelding shows a steady, progressive increase in liability with AGE and CMD.

**Figure 4 animals-11-01028-f004:**
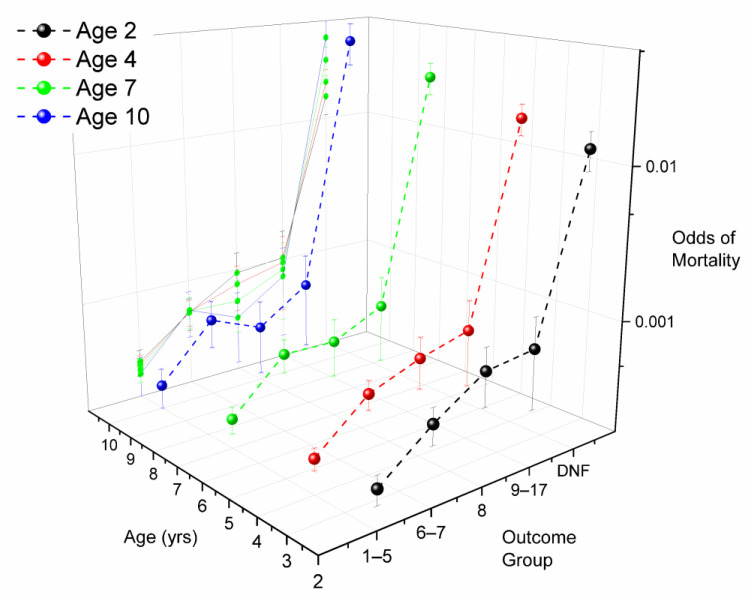
Mortality data from the Ontario Death Registry for work-events involving standardbred horses racing in the Province of Ontario from 2003–2015. Odds of mortality (95% confidence intervals for the mean, unit of interest-work-event), by work-event outcome for AGE groups 2, 4, 7 and 10, describing an AGE×OUTC (outcome) interaction identified through logistic regression analysis. Population-all work-events except scratches in the study period (*n* = 1,713,225). Mortality odds are stratified by outcome. Note the tendency for mortality within AGE to increase as finishing position falls back. DNF outcomes consistently have higher mortality odds than all other finishing positions, finish group 1–5 consistently has the lowest odds.

**Figure 5 animals-11-01028-f005:**
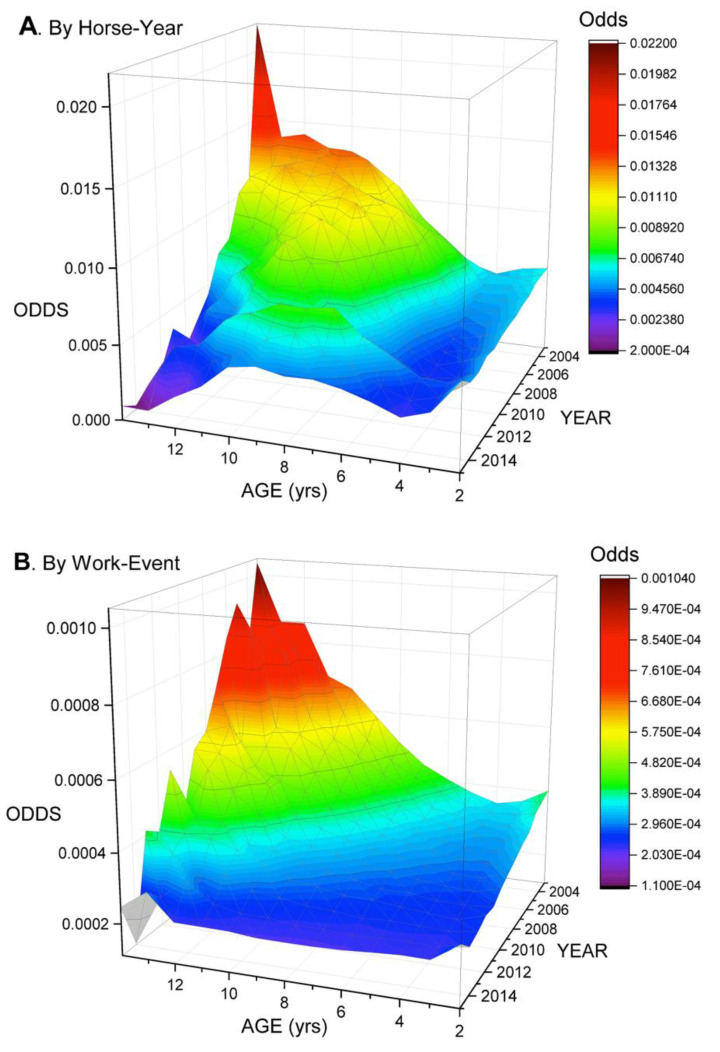
Odds of mortality, by YEAR and AGE, for standardbred horses racing in the Province of Ontario from 2003–2015 and describing an AGE×YEAR interaction identified through logistic regression analysis of mortality data from the Ontario Death Registry. (**A**) Unit of interest-horse-year. Population-all horse-years (*n* = 125,200). When all other effects are held constant, odds of mortality by horse-year decline by YEAR over the study period and show a declining effect of AGE. The greatest fall is for older horses, while mortality odds for younger horses decline far less over the study period. (**B**) Unit of interest-work-event. Reference population-all work-events (*n* = 1,778,330). A similar decline is evident in mortality by work-event. There was no significant change in population distribution by AGE over the study period. Note the difference in scale between the two images, mortality odds by horse-year being 20 times higher than by work-event, a reflection of the number of annual work-events undertaken by horses.

**Figure 6 animals-11-01028-f006:**
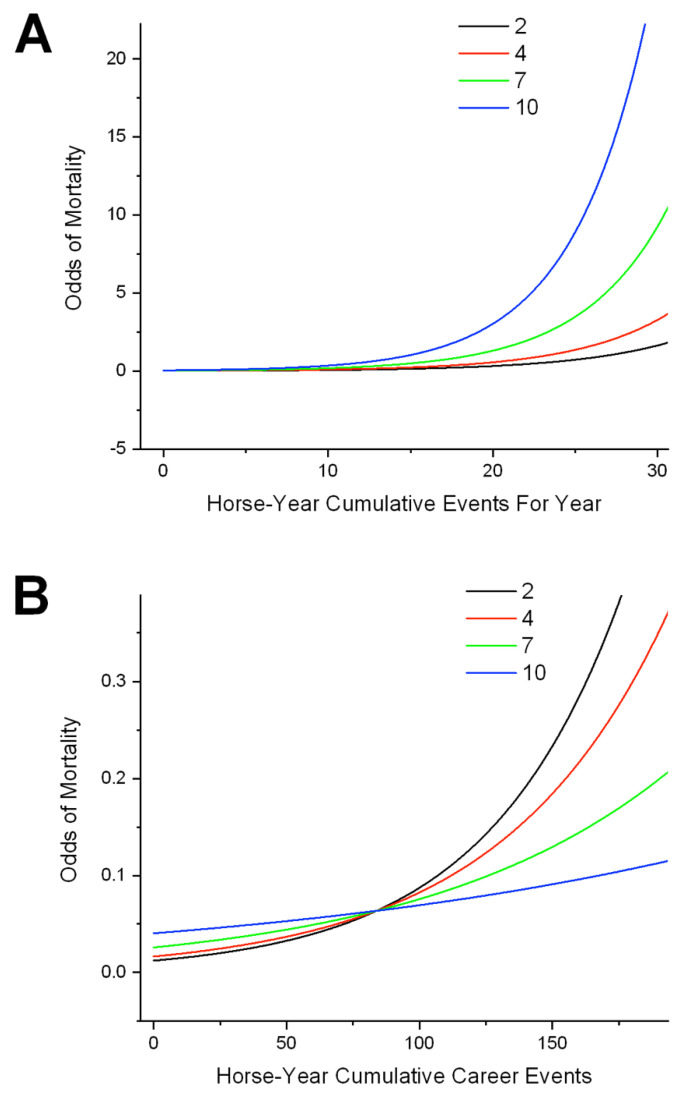
Odds of mortality (statistical mean, unit of interest-horse-year), showing relationships between AGE and indices of cumulative work for standardbred horses racing in the Province of Ontario from 2003–2015, and describing significant interactions identified through logistic regression analysis of mortality data from the Ontario Death Registry. Population—all horse-years for horses racing in the Province in the study period (*n* = 125,200). Maximum cumulative year work-events (CMYR) in this dataset was 57, maximum cumulative career work-events (CMCAR) was 486. Curves are theoretical and describe the trajectory of the relationships identified; many horses, particularly younger animals, would not achieve the number of work-events described. (**A**) AGE×CMYR (cumulative work-events for the year). As AGE rises, the impact of rising CMYR on mortality odds accelerates, suggesting decreasing tolerance of annual workload with increasing age. (**B**) For the interaction AGE×CMCAR at horse-year level, increasing cumulative career work-events has the reverse effect, odds increasing more rapidly for younger than older horses. Trajectories speak to the impact of career intensity. Both sets of relationships are influenced by progressive withdrawal of less robust horses and a survivor effect. Note the differences in scale for the Y axes. Impacts on mortality odds are smaller in B than A, suggesting workloads compressed into shorter periods may be more damaging than equivalent loads that are more spread out.

**Table 1 animals-11-01028-t001:** Presenting and consolidated complaints for standardbred cases in the Ontario Racing Death Registry from 2003–2015. MSI: musculoskeletal injuries.

Presenting	Consolidated	Count	Total	%
Fracture		233		
Catastrophic soft tissue injury		103		
Exertional rhabdomyolysis		3		
Chronic musculoskeletal		7		
	MSI		346	35.6%
Collapse		3		
Dropped Dead		103		
Heart pathology		2		
Found dead		47		
	Died Suddenly		155	15.9%
Colic		154		
	Colic		154	15.8%
Medical complaint		71		
Laminitis		3		
Diarrhea		20		
Respiratory problem		19		
Neoplasia		2		
Bacterial infection		1		
	Medical		116	11.9%
Septic arthritis		29		
Medication reaction		52		
Scrotal hernia		7		
Phlebitis		1		
	Iatrogenic		89	9.1%
Self-inflicted trauma		15		
Off-Track accident		37		
On-track accident		23		
	Accidents		75	7.7%
Neurological		32		
	Neurological		32	3.3%
Epistaxis		2		
Severe hemorrhage		3		
	Hemorrhage		5	0.5%
Unknown		1		
	Unknown		1	0.1%
Total		973	973	100.0%

**Table 2 animals-11-01028-t002:** Glossary of variables, terms, and abbreviations.

Variable	Definition	Range	Type
AGE	Age in years	2–16	Continuous
CMCAR	Cumulative career work-events*	1–486	Continuous
CMD	Cumulative days in racing in current season *	1–366	Continuous
CMYR	Cumulative work-events in current season *	1–57	Continuous
DAY	Day of the week	1–7	Categorical
DBD	Days between work-event and death for registry cases	0–60	Continuous
DOB	Calendar day of birth	1–366	Continuous
DR	Death Registry status	0-not in Registry, 1-in Registry	Binary
FPOS	Finish positions for work-event	1–5 (positions 1–5), 6–7 (positions 6–7), 8 (position 8), 9–17 (positions 9–17)	Categorical
GAIT	Gait for work-event	Trotter or Pacer	Categorical
OUTC	Outcome for work-event, including finish position, SCRATCH or DNF	1–5 (positions 1–5), 6–7 (positions 6–7), 8 (position 8), 9–17 (positions 9–17), SCRATCH, DNF	Categorical
PPOSN	Post position for work-event	1-17	Continuous
SEX	Sex at time of work-event	F-female, G-gelding, S-Stallion	Categorical
START	Type of work-event	Y/N, Race or non-race	Categorical
TATTOO	Unique horse identifier		String
TC	Track Class, surrogate measure of caliber of competition	A, B, C	Categorical
TRACK	Track where work-event took place		String
YD	Work-event date converted to calendar day of year or yearday	1–366	Continuous
YEAR	Calendar year for work-event	2003–2015	Continuous
YOB	Horse year of birth	1989–2013	Continuous

SCRATCH—withdrawn before work-event start; DNF—did not finish work-event; MSI—musculoskeletal injury; CI—confidence interval. *—including current work event.

**Table 3 animals-11-01028-t003:** Results of Logistic Regression Modelling of Associations with Membership in the Ontario Racing Death Registry (binary response) for Standardbred Work-events for the period 2003–2015—By work-event Outcome.

	COMPLETED Work-Events					DNF Work-Events					SCRATCH Work-Events				
Work-Events	1,706,499					6726					65,105				
Mortalities	575					207					191				
Variable	Est. †	s.e.	*p*-Value	OR	95% CI	Est. †	s.e.	*p*-Value	OR	95%CI	Est. ^†^	s.e.	*p*-Value	OR	95%CI
Intercept	−4.1648	0.3848	<0.0001			−4.0115	0.3485	<0.0001			−4.5684	0.2365	<0.0001		
AGE (yrs, 4.91)	−0.3710	0.0852	<0.0001	0.690	0.584–0.815	0.1799	0.0313	<0.0001	1.197	1.126–1.273			n/s		
SEX (F vs S)	−2.1100	0.4514	<0.0001	0.121	0.050–0.294	−0.9446	0.2151	<0.0001	0.389	0.255–0.593	−1.1958	0.2814	<0.0001	0.302	0.174–0.525
SEX (G vs S)	−3.4265	0.4234	<0.0001	0.024	0.005–0.130	−0.9323	0.1916	<0.0001	0.394	0.270–0.573	−1.1509	0.2728	<0.0001	0.316	0.185–0.540
GAIT (P vs T)			n/s			0.6498	0.1874	0.0005	1.915	1.326–2.765			n/s		
START (N vs Y)	0.3337	0.1315	0.01	1.396	1.079–1.807	−1.3341	0.2200	<0.0001	0.263	0.171–0.405			n/s		
OUTC (1–5 vs 9–17)	−1.2082	0.1375	<0.0001	0.299	0.228–0.391			N.A.					N.A.		
OUTC (6–7 vs 9–17)	−0.6248	0.1442	<0.0001	0.535	0.404–0.710			N.A.					N.A.		
OUTC (8 vs 9–17)	−0.3626	0.1638	0.03	0.696	0.505–0.959			N.A.					N.A.		
YEAR (5.28)	−0.0479	0.0122	<0.0001	0.953	0.931–0.976			n/s			−0.0546	0.0228	0.02	0.947	0.905–0.990
TC (A vs C)	−0.4516	0.1535	0.003	0.637	0.471–0.860	0.7131	0.2624	0.007	2.040	1.220–3.412			n/s		
TC (B vs C)	−0.2410	0.1262	0.06	0.786	0.614–1.006			n/s					n/s		
CMCAR (/10, 4.38)			n/s					n/s			−0.0792	0.0495	0.1	0.924	0.838–1.018
CMD (/10, 123.89)	−0.0997	0.0276	0.0003	0.905	0.857–0.955			n/s					n/s		
CMCAR × SEX (F vs S)			n/s					n/s			0.1259	0.0561	0.02	1.134	1.016–1.266
AGE × SEX (F vs S)	0.2369	0.1098	0.03	1.267	1.022–1.572			n/s					n/s		
AGE × SEX (G vs S)	0.4770	0.0942	<0.0001	1.611	1.340–1.938			n/s					n/s		
CMD × SEX (G vs S)	0.1128	0.0316	0.0004	1.119	1.052–1.191			n/s					n/s		
CMD × AGE	0.0167	0.0052	0.001					n/s					n/s		
CMD × AGE × SEX (G)	−0.0163	0.0057	0.004					n/s					n/s		

^†^ Estimate, GAIT: P—Pacer, T—Trotter; SEX: F—female, G—gelding, S—stallion; YEAR—calendar year, 0–12 (2003–2015); AGE in years; START: N—qualifier or schooling race, Y—race start; TC—track class, A—C; OUTC—work-event outcome, 1–5—finished in the first 5; 6–7–finished 6th or 7th; 8—finished 8th; 9–17—finished 9th to 17th; DNF—Did Not Finish; SCR scratched; CMD—cumulative days of racing for the current year, in increments of 10; CMCAR - cumulative work-events for career to current year, in increments of 10; N.A.—not applicable; n/s—not significant. The table shows results significant at *p* < 0.05. s.e.—standard error; OR—odds ratio; CI—confidence interval. Referents for categorical variables and means for continuous variables are underlined. YEAR was treated as continuous in these analyses.

**Table 4 animals-11-01028-t004:** Populations at Risk, Mortality Rates, and Common Presenting Complaints by Outcome and Track Class for Standardbred Racehorse Work-events in the Province of Ontario, 2003–2015.

Work-Event Level	Pop.n at Risk *	Track Class		
		A	B	C
Distribution by Class, %	1,778,330	21.35	67.91	10.74
		Track Proportion **		
COMPLETED Outcomes	1,778,330	96.92	95.73	95.49
DNF Outcomes	1,778,330	0.24	0.40	0.54
SCRATCH Outcomes	1,778,330	2.84	3.87	3.97
		Mortality Rates ^†^		
By Total Work-events	1,778,330	0.472	0.548	0.691
COMPLETED, START = Y	1,514,996	0.296	0.325	0.417
COMPLETED, START = N	191,503	0.359	0.416	0.416
DNF Total	6,726	36.876	29.954	29.126
DNF, START = Y	4,178	71.759	42.0	33.512
DNF, START = N	2,548	6.122	9.582	17.606
SCRATCH, START = Y	65,105	2.973	2.844	3.432
By Common Complaint:		Mortality Rates ^†^		
COMPLETED MSI	1,706,499	0.087	0.086	0.105
COMPLETED D.S.	1,706,499	0.047	0.053	0.058
COMPLETED Accident	1,706,499	0.016	0.023	0.037
COMPLETED Iatrogenic	1,706,499	0.037	0.031	0.047
DNF MSI	6,726	27.115	21.575	24.272
DNF D.S.	6,726	6.511	5.072	2.913
DNF Accident	6,726	3.254	1.885	0.971
DNF Iatrogenic	6,726	0	0.209	0.971
SCRATCH MSI	65,105	0.464	0.577	0.528
SCRATCH D.S.	65,105	0.836	0.342	0.528
SCRATCH Accident	65,105	0.186	0.385	0.132
SCRATCH Iatrogenic	65,105	0.929	0.278	0.396

* size of population at risk (work-events, all tracks); ** percent of track class-specific outcomes; ^†^ rate per 1000 track-specific outcomes. MSI—musculoskeletal injury; D.S.—died suddenly; DNF—did not finish.

**Table 5 animals-11-01028-t005:** Results of Logistic Regression Modelling of Associations with Membership in the Ontario Racing Death Registry (binary response) for Standardbred Horses for the period 2003–2015, Unit of interest—Horse-Year.

Horse Years	125,200	s.e.	*p*-Value	OR	95% CI
Mortalities	973				
Variable	Estimate				
Intercept	−4.6994	0.3405	<0.0001		
AGE (yrs, 4.57)	0.1492	0.0902	0.0981	1.161	0.973–1.385
SEX, F vs. S	−1.1493	0.2261	<0.0001	0.317	0.203–0.494
SEX, G vs. S	−1.6949	0.2233	<0.0001	0.184	0.119–0.284
TC, A vs. C	−0.2281	0.2135	0.2852	0.796	0.524–1.210
TC, B vs. C	0.1298	0.1879	0.4895	1.139	0.788–1.646
CMYR (14.20)	1.5010	0.0190	<0.0001	4.486	3.091–6.510
CMCAR (37.07)	0.2320	0.0390	<0.0001	1.261	1.168–1.361
CMD (163.23)	−0.0608	0.0130	<0.0001	0.941	0.917–0.965
YEAR (5.24)	0.0266	0.0264	0.3142	1.027	0.975–1.082
AGE × SEX, F vs. S	0.0827	0.0473	0.0808	1.086	0.990–1.192
AGE × SEX, G vs. S	0.2110	0.0429	<0.0001	1.235	1.135–1.343
CMD × TC, A vs. C	−0.0185	0.0130	0.1547	0.982	0.982–1.007
CMD × TC, B vs. C	−0.0380	0.0110	0.0005	0.963	0.942–0.984
AGE × CMYR	0.0657	0.0191	0.0006		
AGE × CMCAR	−0.0178	0.0048	0.0002		
AGE × YEAR	−0.0145	0.0051	0.0050		
CMYR × YEAR	−0.0239	0.0115	0.0370		
AGE × AGE	−0.0168	0.0068	0.0132		
CMYR × CMYR	−0.4000	0.0374	<0.0001		

OR—odds ratio; CI—confidence interval; GAIT: P—Pacer, T—Trotter; SEX: F—female, G—gelding, S—stallion; YEAR—calendar year, 0–12 (2003–2015); AGE in years; START—N-qualifier or schooling race, Y-race start; TC—track class, A—Premier, B—Signature, C—Grassroots and Regional; CMYR—cumulative work-events for the current year, in increments of 1; CMD—cumulative days of racing for the current year, in increments of 1; CMCAR—cumulative work-events for career to current year, in increments of 1; The table shows results significant at *p* < 0.05 unless involved in an interaction. Referents for categorical variables and means for continuous variables are underlined.

## Data Availability

Mortality data are the property of the Alcohol and Gaming Commission of Ontario, Regulatory Compliance Branch, to whom requests should be directed. standardbred performance data are the property of Standardbred Canada, to whom requests should be directed.

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
