# Peer review of "Factors Associated with Mortality in Ontario Standardbred Racing: 2003–2015"

_animals, 2021, doi:10.3390/ani11041028_

Round 1
Reviewer 1 Report
The original article about “ Risk factors for Horse Mortality in Ontario Standardbred Racing 2003-2015” is of great interest for trainers, owners and veterinary practitioners and enthusiasts of horse racing. In addition, there is only several studies connected with Standardbred. The present paper is interesting and well written. However, it needs some correction.
First of all the Simple Summary and Abstract is too long. Only brief and the most important information should be mentioned. In addition, Animals journal required 200 words if I am correct. Thus, I encourage Authors to shorten this parts.
Introduction
In some parts there are to many information, whereas more important are missing. I encourage to re-written a little bit this part.
L56 – information connected with the welfare of the race horses should be added. It is really important that not only injury and fatality in race horses are important. Thus, information about the stress response after race training and competition in horses should be added.
L76-79 – more clarifications are missing. It is important to mention how age, sex, training influence on morbidity and mortality.
L80 – more information about management leading to the decrease of stress response in race training should be added.
L85 – information about prevention is missing. The most valuable parameter in monitoring the training progress is blood lactate concentration. In recent years, also other very novel techniques are used, ex. infrared thermography which also correlates with lactate concentration. Also other blood parameters may be used for training monitoring such as changes in PBMCs proliferation and activity, cortisol concentration or cytokines mRNA expression. Thus, I encourage the Authors to add short paragraph connected with those techniques which allow to prevent the race horses overload and then injury.
L86-93 – this part should be in materials & methods
L95-102 – this part should be shorten.
L104-115 – the aim of the study should be more precise because it is confusing.
Materials & Methods
L126- in my opinion the nine groups should be added there, not only cited from previous Authors work.
Results
The results are clearly presented, however Figure 1 and 2 should be changed because they are really small and of low quality so it is hard to read.
Discussion
This part is clear and well written. However, some clarification are needed.
L525 – this sentence should be clarified how age and sex influenced in race horses from cited publications and how it differs from present study results.
L538 – the environmental factors should be mentioned as well as the stress response after race training and competition in horses.
L551 – the information about pasture time influence should be added.
L565 – maybe it is connected with overtraining?
L569 – probably higher winnings and prizes of the horses are connected with that.
L571 – the A,B,C should be explained.
Limitations
Unpredictable things which are not correlated with training regimen such as aorta aneurysm should be mentioned.
References
Authors should avoid publication older than 30 years.
Reviewer 2 Report
The aim of this study was to determine risk factors for horse mortality in Ontario Standardbread racing horses, using data from 2003 to 2015.
It is very much appreciated to read manuscripts with this level of analysis.
Some minor comments/suggestions are added in the .pdf file attached.
Two specific comments can be highlighted:
- How were the models compared? using AIC, BIC, LRT or any other, think that it is important to state that in the M&M section.
- Some parts of the introduction (particularly at the end of it) seem to be more connected with a M&M section and with Discussion and Conclusion sections, so those should be removed from the Intro and placed elsewhere.

Reviewer 3 Report
This manuscript describes the most important risk factors for increased mortality in standardbred race horses. The context and results of the study are presented very well. The manuscript will be a highly useful reference for the standardbred racing industry. I therefore recommend to publish this manuscript with minor revisions.
The written language was very good, I have only minor comments.
Page 2, line 61-62: I do not understand this statement. "in the context of racing" is very vague. Do you mean during racing and training or do you mean something else? Please revise.
Page 4, table 1: DR variable is binary (not biinary)
Page 14, line 378: this is the first time you use the term iatrogenic mortality. Could you please explain somewhere what you mean by this term specifically?
Reviewer 4 Report
Dear Authors
Thank you very much for this very interesting, well presented, well written article on the causes of mortality in racehorses.
I don't have many comments.
There are a lot of figures but if it is in agreement with the magazine it is rather an advantage because the subject lends itself well to it.
I am not sufficiently qualified in statistics to ensure the quality but the treatment of the data and the exposure of the figures seems to me to be appropriate. There are a lot of abbreviations - the lexicon is a plus but the text should be read without referring to them too much. In the result section there are sentences which are more a matter of interpretation and should be discussed (e.g. £222, 267). I hope I didn't miss any information, but I would have liked to know what the difference in risk is between flat, steeplechase and cross-country running at full gallop. I do guard duty at the racecourses and the jumping races seem to be much more risky than the others, to the point of wondering if it is still ethical.
Sincerly yours
Reviewer 5 Report
The group selection and the applied statistical methods correspond to the purpose of the work. A large number of cases provide good material for further research and other research groups. I recommend publishing the work without corrections.
